# Fly Ash from Lignite Combustion as a Filler for Rubber Mixes. Part I: Physical Valorization of Fly Ash

**DOI:** 10.3390/ma15144869

**Published:** 2022-07-13

**Authors:** Wojciech Orczykowski, Dariusz M. Bieliński, Rafał Anyszka, Zbigniew Pędzich

**Affiliations:** 1BESTGUM POLSKA Ltd., Św. Barbary 3, 97-427 Rogowiec, Poland; wojciech.orczykowski@bestgum.pl; 2Institute of Polymer & Dye Technology, Lodz University of Technology, Stefanowskiego 16, 90-537 Lodz, Poland; rafal.anyszka@p.lodz.pl; 3Department of Ceramics and Refractories, AGH University of Science and Technology, Mickiewicza 30, 30-059 Cracow, Poland; pedzich@agh.edu.pl

**Keywords:** fly ash, fractionation, characterization, rubber vulcanizates, mechanical properties

## Abstract

The potential use of fly ash (FA) originating from lignite combustion at the Belchatow Power Plant (Poland) as filler for rubber mixes was evaluated. Samples of fly ash collected from heaps created in different years were compared according to their chemical and phase composition, particle size distribution, and morphology. The sieve fractionation of fly ash results in size fractions of different chemical structures, phase compositions, and morphologies, reflected in changes to their specific surface area, surface energy, and activity in rubber mixes. Fractionation turned out to be more effective than grinding from the point of view of using ash as a filler for rubber mixes, because it results in higher specific surface area (SSA) and chemical composition differentiation. Carbon black can be replaced by up to 40% by weight with the fly ash fraction (FFA) of dimensions below 125 µm, without any significant deterioration in the mechanical properties of styrene butadiene rubber (SBR) vulcanizates filled with 50 phr of active carbon black (N 220). Despite the larger fly ash fraction of grain dimensions in the range between 125 and 250 µm presenting the highest specific surface area, the particle size adversely affects its strengthening effect in rubber. Taking into account all the tests performed, ranging from morphology, Payne effect and bound rubber, to mechanical and abrasion tests, the highest potential effectivity is presented by a sample containing 30 phr of N 220 and 20 phr of FFA of grain sizes from 63 to 125 µm. The obtained results indicate that fractionation seems to be an effective physical method of fly ash valorization.

## 1. Introduction

Despite the constant increase in environmental and ecological limitations, the scale of using lignite (brown coal) as fuel and the waste generated as a result of its combustion in conventional boilers are still amongst the main ecological problems facing the energy and mining industry in Poland. Apart from a number of other wastes, ash in the amount of approx. 4 million Mg/year is produced as a result of lignite combustion [1]. The annual production of coal ash worldwide is estimated at around 700 million tons. The utilization of fly ash in construction, as cement addition, in precast elements, in roller compacted concrete (RCC), in road sub-base, as lightweight aggregate, and, finally, as an adsorbent for the removal of some compounds, has been reported [2]. Around the world, coal represents 34% of power generation, and this figure is expected to be reduced to 10% by 2040 [3]. Many studies [4,5,6] present methods of fly ash management, including the production of building materials, cement, road construction, and mines. Fly ash from the Belchatow Power Plant has been repeatedly tested by various authors in terms of its quantitative and qualitative composition [7,8,9,10,11]. Despite the fact that this ash is generally aluminosilicate–calcareous, the presented results of analyses of its chemical composition differ significantly in the content of silicon, alumina, and calcium oxide (Table 1). This can be the result of the different periods in which the samples were collected [11].

Fly ash, in addition to minerals, also contains coal that is not completely burnt [12]. As it turns out, the main quality parameters of lignite mined in the Belchatow mine also have a direct impact on the variety of combustion products, including fly ash [13]. It is the type of burnt coal that shapes their properties and, depending on the location of extraction, a large variation in the chemical and phase composition of the ash is observed [7]. The factors that additionally determine the ash composition are the combustion technology and temperature, as well as the flue gas desulphurization method [14]. Generally, radioactivity is one of the most serious problems when using fly ash as a building component. However, Garbacik and Baran [7] examined the natural radioactivity coefficients of fly ash from the Belchatow Power Plant (Poland), which do not pose a threat to human health.

The possibility of managing fly ash in the form of an additive to plastics invariably arouses great interest. Despite many efforts, often successful [15,16], the problem concerning the deterioration of the mechanical properties of the materials, in general, still remains unresolved. In its native form, fly ash presents low potential as an active filler for polymers due to its heterogeneous composition, big grain size, and legal restrictions on processing. This is why there is a limited number of technologies using fly ash on a large scale, thus creating new landfills. The case is even worse when it comes to rubber compounds. Researchers, however, more and more often use fly ash from the combustion of coal, trying to use it as a filler and examining its influence on the properties of rubber composites. Attempts to add fly ash to mixtures based on styrene–butadiene rubber (SBR) were carried out mainly in order to reduce costs and improve the strength of the vulcanizates [17]. It was also investigated how the content of metal ions of variable valence, present in the ash, affects the storage stability of the mixtures and the degradation process of their vulcanizates. The obtained results confirm the necessity to add anti-aging substances to avoid the accelerated aging of rubber products containing the ash [18].

After positive results were derived from introducing native fly ash to rubber mixtures based on various rubbers, both as the sole and additional filler, attempts were made to valorize it by various types of physical and chemical modifications. An example of physical modification includes grinding the ash and using it as an additive to mixtures used in the production of tires [19]. The ash was also divided into fractions of various grain sizes [20]. Fineness is a key parameter in this material. Therefore, the radiological risk should be mentioned [21]. The filler prepared in this way was introduced into the rubber mixtures and their properties were tested. An improvement in the physical properties, mainly the mechanical strength, of the vulcanizates was noticed along with the decreasing size of the ash grains. An improvement in the mechanical properties of the ash-filled vulcanizates can also be obtained by selecting an appropriate coupling agent for the ash particles to interact with in the rubber, e.g., by means of silane treatment [22]. 

The aim of this study was to investigate:The composition and properties of fly ash originating from Belchatow Power Plant (Poland), and its fractions in terms of its applicability as filler for rubber mixes;The influence of the content of fly ash or its fractions in the rubber mixes on selected mechanical properties of the vulcanizates by partially replacing technical carbon black.

## 2. Materials and Methods

### 2.1. Materials

#### 2.1.1. Fly Ashes and Their Grinding

The subject of the research was fly ash, constituting the furnace waste from lignite combustion in the Belchatow Power Plant, marked as follows:−A 2017—fly ash sample collected in 2017;−B 2018—fly ash sample collected in 2018 (data available in Appendix A).

The ash samples were ground in a prototype 1-L laboratory rotary vibration mill with grinders 5 mm in diameter, made of ZrO_2_ (own construction). The ratio of the mass of grinding balls to the mass of the ground material was 10:1. Milling took place in distilled water, with a rotational speed of 400 rpm and a vibration frequency of 60 Hz, for 4 h.

#### 2.1.2. Rubber Compounds and Their Vulcanizates

The rubber mixes were prepared in two stages. In the first stage, the ingredients were mixed in a 2.5 L chamber in a 03 551 Buzuluk (Czech Republic) laboratory internal mixer, under the following conditions:Rotor speed—100 rpm;Chamber fill factor—75%.

The sequence of mixing, time, and the thermal conditions of the mixing stages are summarized in Table 2.

At each stage, the ram was lowered and raised in order to minimize dead zones during mixing. In the second stage, a crosslinking unit (sulfur and accelerators) was added to the rubber mix on a WT300 FAMPA (Jelenia Góra, Poland) laboratory two-roll mill and mixed during ca. 8 min. Table 3 contains the composition of rubber mixes with fly ashes, sampled in 2017 (A), partially replacing N 220 carbon black (CB).

The fly ash was sieved with a sieving machine (MULTISERW-Morek Ltd., Marcyporęba, Poland), and filler fractions with four different ranges were prepared: FFA < 0.063, 0.063 < FFA < 0.125, 0.125 < FFA < 0.250 mm. One more without sieving (native fly ash named as FA-All) was also prepared; all are designated according to Table 4.

Based on mechanical strength examinations, further rubber compounds, containing 20 phr of fly ash of 3 lower size fractions (FFA-A), and completed to 50 phr of the total filler loading with 30 phr of CB (Table 5), were tested. This guaranteed the reasonable mechanical properties of the rubber vulcanizates. Due to the unacceptable dispersion of the biggest FA fraction (particles of average dimension >0.250 mm, pointing to poor mechanical properties in the rubber vulcanizates), rubber compounds with this addition were not made. BET measurements revealed a higher specific surface area in the fractions originating from FA-A (Table 12), which results in a better strengthening effect in comparison to the FA-B used as the CB co-filler in rubber vulcanizates. Further studies have been limited to the formulation based on fly ash A (Table 5). The idea behind the work was that we would first compare vulcanizates with different ashes (A and B) and create mixtures in different CB/FA ratios. The results of the vulcanization kinetics show that for the S + CBS system, very long t_2_ and t_90_ times were obtained. Therefore, it was decided to add TMTD to reduce vulcanization time, making the process commercially viable.

The Mooney viscosity of rubber compounds, ML(1 + 4) at 100 °C, was determined with an MV 2000 Mooney viscometer (TA Instruments, New Castle, DE, USA), according to PN-ISO 289-1. Rubber samples were steel-molded at 145 °C during an optimum vulcanization time of t_90_, determined vulcametrically by an MDR 3000 oscillating disk rheometer from Alpha Technologies (Wilmington, DE, USA), according to PN-ISO 3417.

### 2.2. Methods

#### 2.2.1. Size Distribution Analysis of Fly Ash Particles 

A Mastersizer 2000 with Hydro S attachment (Malvern Instr., Malvern, UK) was used. The particle size distribution of the powders was determined by laser light diffraction (LD). The device was equipped with a red laser (633 nm, HeNe gas laser) and a blue laser (466 nm, LED laser) for measuring the smallest particles (size range from 100 nm to 1 mm).

The software analyzed the diffraction pattern of the powder particles based on the Fraunhofer scattering theory or the Mie theory [23]. A water suspension of powders containing a small amount of a dispersing agent, e.g., Dispex (BASF, Ludwigshafen, Germany), was prepared with the use of high-energy ultrasounds and then fed to the measuring cell. During the measurement, the suspension was in constant motion, forced by the agitator located in the adapter, and flowed through the measuring cell many times. Such dynamic measurement conditions made it possible to record the diffraction pattern of a large number of particles. 

#### 2.2.2. Fractionation of Fly Ash Particles

Fractionation of the fly ashes: A 2017 and B 2018 were performed using a laboratory sieve shaker LPzE-3e (MULTISERW-Morek Ltd., Marcyporęba, Poland), operating with 0.250, 0.125, and 0.063 mm sieves. Experimental conditions applied: test time—3 min, amplitude—1.5 mm, vibration frequency—50 Hz. The process was carried out by sieving the ash until a constant mass was obtained on each sieve. The measurement error was calculated using the student’s *t*-test for α = 0.005. 

#### 2.2.3. Surface Energy of Fly Ash Particles

The surface free energy (SFE), and its components, of the fly ashes were determined by a K100 MKII tensiometer (KRÜSS GmbH, Hamburg, Germany). Capillary constant was determined with heptane. Other solvents applied to designate the SFE of the fly ash particles were 1,4-dioxane and methanol. SFE and its dispersive and polar parts were calculated by applying the approach proposed by Owens–Wendt–Rabel–Kaelble (OWRK) [24].

#### 2.2.4. Specific Surface Area of Fly Ash Particles

The specific surface area (SSA) of the fly ash and its fractions were determined with an ASAP 2010 (Micromeritics, Norcross, GA, USA) instrument, applying the Brunauer–Emmet–Teller (BET) equation. The samples were degassed at 160 °C for 24 h, and the obtained vacuum value was 2 µTr. The measurement was carried out at the temperature of liquid nitrogen. Specific Surface Area (SSA) was determined utilizing a 5-point BET procedure, according to ASTM D3037 and ASTM D4820 standards. The measurement error was calculated using the student’s *t*-test for α = 0.005. For the comparison, the specific surface areas of fly ashes and their fractions were also calculated from the adsorption of cetyltrimethylammonium bromide (CTAB) measurement, which is believed to better simulate the accommodation of polymer macromolecules on a filler surface [25,26]. The methodology of CTAB has been described in the literature [27,28]. N330 carbon black, giving typical SSA values of 83 m^2^/g, was used for the calibration of the measurements. 

#### 2.2.5. Morphology and Phase Composition of Fly Ash Particles

The morphological analysis of the ashes and their fractions was carried out with the use of a scanning electron microscope (SEM) Nova Nano Sem 200 f (FEI, Altrincham, UK) in combination with an energy dispersive X-ray Spectrometer (EDS), which enabled the analysis of the chemical composition. Observations of the microstructure were carried out on samples covered with a thin layer of carbon, using a reflected electron (BSE) detector with an accelerating voltage of 18 kV, operating under high vacuum conditions, or a secondary electron detector (LVD) with an accelerating voltage of 10–18 kV, operating under low vacuum (60 Pa).

#### 2.2.6. XRD Analysis of Fly Ash 

The phase analysis of fly ashes and their fractions was carried out with an Empyrean diffractometer (Malvern Panalytical, Westborough, MA, USA), operating with a monochromatic beam of a wavelength of the Cu Kα emission line. Measurements were taken over an angular range of 5–90° on a 2θ scale, applying a step of 0.008°. Qualitative and quantitative analyses (Rietveld’s method) of the phase composition were performed using a 3.05 version of X’Pert HighScore Plus software, (PANalytical BV, Malvern, UK) based on the following databases: PDF-2 (2004), ICSD Database FIZ Karlsruhe (2012).

#### 2.2.7. Filler Dispersion 

The filler dispersion of rubber vulcanizates was determined using a DisperTester 3000 instrument (Montech, Columbia City, IN, USA), according to ISO 11345:2006. Filler dispersion (D), denoting the degree of filler dispersion in the vulcanizate, is described by the formula:D = (1 − N_a_/N_tot_) × 100%
where:N_a_—total number of pixels containing agglomerates (white areas);N_tot_—total number of pixels in the image.

Data processing was carried out with the help of MonDispersion software. The magnification applied was ×1000, with all the images referring to an area of 250 × 150 µm.

#### 2.2.8. Bound Rubber Content (BdR)

The bound rubber content of the rubber compounds was determined by extracting the unbound components, such as free rubber and other ingredients, with toluene at room temperature for 6 days, and afterwards with n-hexane at room temperature for 1 day. Then the samples were dried at room temperature for 2 days. The weights of the samples before and after the extraction were measured and the bound rubber contents were calculated with the equation:(1)$$BdR\;\left(\%\right)=100\times \{W_{fg}-W_{t}\left[\frac{m_{f}}{\left(m_{f}+m_{r}\right)}\right]/\{W_{t}\left[m_{r}/\left(m_{f}+m_{r}\right)\right]\}$$
where *BdR* is the bound rubber content, *W_fg_* is the weight of filler and gel, *W_t_* is the weight of the sample, *m_f_* is the fraction of the filler in the compound, and m_r_ is the fraction of the rubber in the compound [29]. The procedure of extracting the rubber component (unbound rubber) from the compound was as follows: (1) unbound rubber chains were extracted from the compound with toluene at room temperature, (2) ethanol was gradually added to coagulate the polymer component, and (3) the coagulated rubber was washed with ethanol and dried in a vacuum oven.

#### 2.2.9. Payne Effect

The Payne effect of the unvulcanized rubber compounds was examined using a MonTech D-RPA 3000 (Buchen, Germany), operating with the strain sweep ranging from 0.5 to 100% at 100 °C. The Payne effect of the rubber samples was determined as the difference between storage moduli at the highest (100%) and the lowest (0.56%) deformations.

#### 2.2.10. Mechanical Properties of Rubber Vulcanizates

The rubber vulcanizates containing fly ashes or their fractions, subjected to static elongation, were examined with a 1435 universal mechanical testing machine operating with an optical extensometer (Zwick/Roell, Ulm, Germany), according to PN-ISO 37; 2.0 ± 0.2 mm-thick and 25 mm measuring distance dumb-bell specimens were elongated by 500 mm/min. Six samples per material were tested and the experimental values averaged. Stress at elongation of 100, 200 and 300% (SE 100, SE 200 and SE 300, respectively), the tensile strength (TS), as well as the elongation at break (Eb) were determined. The hardness of rubber vulcanizates was determined with a Zwick Shore A durometer 3130 (Ulm, Germany), according to PN-EN ISO 868. The measurement error was in each case calculated using the student’s *t*-test for α = 0.005.

#### 2.2.11. Abrasion Resistance of Rubber Vulcanizates

The abrasion resistance of rubber vulcanizates was determined with a SchopperScholbach instrument (VEB Thuringer Industriewerk Raunstein, Dresden, Germany), according PN-ISO 4649: 2007, met. A.

## 3. Results and Discussion

### 3.1. Analysis of Fly Ash and Its Fractions

#### 3.1.1. Particle Size Distribution

The performed particle size analysis of native fly ashes confirmed that the basic grain fraction of the tested ashes was the clay–dust fraction (from <0.002 to 0.05 mm), constituting approx. 60% of the sample mass [30]. The rest, approx. 40%, was the sand fraction (from 0.05 to 2 mm). There were no significant differences between the two tested ash samples from the two consecutive years.

The particle size distribution for ash samples from 2017 (a) and 2018 (b), before grinding (designated as FA (Figure 1A and Appendix A)) and after grinding (as FFA (Figure 1 and Appendix A)), is presented below. The charts in the individual pairs are very similar. For unmodified samples, the dominant fraction of grains occurred at a size of approx. 100 µm, which is a slightly higher value than presented in the literature [31]. As expected, further grinding caused the main fraction size to decrease up to approx. 10 µm. The 2018 ash sample was characterized by a higher amount of larger grain sizes, which may indicate that the grinding process of this batch may require longer to obtain the same particle size distribution as for the 2017 sample.

Based on the results obtained, FA from 2017 (A) was selected for further studies.

#### 3.1.2. Morphology

The SEM EDS study was carried out in order to perform a qualitative analysis of the morphology of the samples A (2017) and B (2018) before and after grinding. Figure 2 and Appendix A show the appearances of the ash samples at different magnifications, so that their shape and the distribution of individual grains are visible. One can see the similarity in the sizes of individual grains and the degree of distribution of individual elements. However, at higher magnifications, the particle size distributions are similar, but their morphologies, despite similar dimensions, differ significantly. This is probably due to the difference in the chemical/phase composition of the ash samples acquired in different years, as discussed later.

After the grinding process, the grain size reduction was clearly visible in both the first (Figure 2) and the second sample (Appendix A). However, with a higher magnification, one can see that there are slightly larger grains in sample B. Both samples contain a lot of irregularly shaped particles, and there are only single spherical grains. The pictures before the grinding modification show more oval objects with rounded edges.

Interesting conclusions are offered by the SEM analysis of fly ash divided into fractions. Due to the fact that the ashes show similar trends with the sizes of individual grains, it was decided to present morphological changes with the example of fly ash A (Figure 3). When comparing different grain sizes, one can see at first glance that as the grain size increases, the morphology of their surface changes. The larger the grains, the more porous their structure becomes. This is of course reflected in the size of the specific surface area, but also, as demonstrated later, in the chemical/phase composition of the individual ash fractions.

#### 3.1.3. Chemical and Phase Composition 

XRD analysis of the virgin ash (FA) composition, prior to grinding, is presented in Table 6 and Appendix A.

On the basis of the X-ray analysis of fly ash samples A (from 2017) and B (from 2018), the minerals included in the ash were identified (Table 7 and Appendix A).

For both samples it can be seen that each of the tested ashes consists mainly of silicon dioxide and aluminosilicates (Table 6 and Appendix A), which is in line with the literature presenting coal fly ash composition from different regions [32]. However, the contents of minerals in the examined ashes are very different from each other (Table 7). Sample A from 2017 shows a higher content of silicon dioxide compared to sample B from 2018, where the akermanite–gehlenite mixture content is the highest (Appendix A). Sample A is more homogeneous in its composition—four types of compounds are visible, with the majority of silicon dioxide content (45.7%). In comparison, sample B consists of as many as seven compounds. In this case, the silicon dioxide content is only 14.3%. This may explain the large difference in the value of the specific surface area (see Table 11 and Appendix A), as well as the surface energy and its components (see Table 13), for the two different ash samples studied. 

Table 8 and Appendix A show the total percentages of oxides and carbon, changing depending on the type of ash and the size of the fraction analyzed after grinding. It is clearly visible that in both cases, with the increase in grain size, the total content of oxides decreases, while the amount of carbon increases.

Table 9 and Appendix A contain the results of qualitative and quantitative “oxide” analyses of fly ashes divided into fractions, based on SEM-EDS data.

Approximately, the crystalline phase constitutes about 50% of the mass of the tested ashes. The weight percentages of the crystalline phases (compared to each other) in the individual ash fractions are presented below. The analysis clearly shows which crystalline phases are fine and which are coarse-crystalline. There are clear changes in the number of individual phases depending on the particle size of a given fraction (Table 10 and Appendix A). 

The most significant change concerns quartz, which is the main crystalline component of the biggest fractions. The same tendency, but on a smaller scale, is observed for mullite and albite, whereas the content of akermanite/gehlenite decreases with the increase in fraction size. 

#### 3.1.4. Specific Surface Area (SSA)

The results presented in Table 11 and Appendix A show that the analyzed samples within the same batch do not differ significantly in terms of the size of the specific surface area, as determined by the BET method. On the other hand, ash samples from 2017 have a specific surface area twice as high as those collected in 2018. This difference (more than twice) was confirmed by SSA measurements using the cetyltrimethyl ammonium bromide (CTAB) method. After grinding, the value of the BET specific surface slightly increased only for FA-A 2017, which confirms that the grinding did not give the desired effect. Therefore, only one CTAB measurement was made for each fly ash sample for comparison purposes. Due to the fact that there was no significant increase in the specific surface area after this process, further measurements were abandoned.

After the ashes were divided into fractions, the BET method was repeated (Table 12 and Appendix A). There is a visible increase in the value of the specific surface area with an increase in grain size. Moreover, when comparing the values for two different ash samples with the same fraction, it is clear that the fractioned ash samples from 2017 have twice the BET value of the fractioned ash samples from 2018. For comparative purposes, the size of the specific surface area was measured using the CTAB method for the sample FA-A from 2017. The aim was to check whether the CTAB results would confirm the previous observations of the dependence of the specific surface area value on the size of the ash grains used for the test. There was also an increase in the CTAB value with an increase in grain size. This confirmed that the increase in the ash grain size for samples from different years affects the increases in BET and CTAB values, so the CTAB study for sample FB from 2018, of lower specific surface area, was again skipped. 

#### 3.1.5. Surface Energy

The surface energies of the ash samples before grinding are compared in Table 13 and Appendix A.

Again, the sample FA-A from 2017 demonstrated higher surface energy, for both dispersive and polar components. Table 14 presents the results of surface energy analysis only for three fractions of the sieved fly ash FFA-A 2017: A < 0.063, 0.063 < A < 0.125 and 0.125 < A < 0.250 mm. Despite the use of various liquids and many attempts, it was not possible to determine the surface energy for the ash sample with a fraction of the grain size >0.250 mm. However, from the point of view of its application in rubber mixes, the biggest fraction is of no interest due to the potential problem with stress generation and crack propagation by the big particles in rubber vulcanizates. It was decided to limit the study on the influence of the grain size over its surface energy value to fly ash A. It was assumed that the direction of changes would be the same for fly ash B. Based on the surface energy results for the virgin fly ashes, the values for the fractions of fly ash B were expected to be lower in comparison to the adequate fractions of fly ash A. 

When analyzing the obtained data, it can be seen that with the increase in grain size, the value of the dispersive component of surface energy increases from the smallest to the biggest grain size ca. 45%. The polar part also increases, but only up to a certain point. As can be seen for the biggest fraction, the polar part is actually residual. It can be concluded that not only the size of the grains, but most of all the chemical composition, and the differences in the crystalline phase structure, are responsible for the different properties of the ash fractions.

### 3.2. Influence of Fly Ash Addition over Properties of Rubber Vulcanizates

#### 3.2.1. Kinetics of Vulcanization and Processability of Rubber Compounds

Fly ash is not active enough to be used as a primary filler in rubber mixes [19]. Expecting better mechanical parameters of the vulcanizates, part of the fly ash added to the rubber compounds was replaced with carbon black. Because of the better parameters of fly ash A, the influence of the fly ash content on the vulcanization kinetics of rubber compounds filled with FA + CB systems is presented only via the example of FA-A from the 2017 filled samples. The compounds containing ash from 2018 behaved in a similar way. The vulcanization kinetics of rubber compounds filled with N 220 carbon black partially replaced by fly ash FA-A from 2017 are presented in Figure 4.

The adequate parameters of vulcanization are presented in Table 15. 

The scorch time of the rubber compounds was extended as the fly ash content increased. This is probably due to the low activity of the ash as a filler. On the other hand, increasing the amount of fly ash in the sample makes the increase in vulcametric modulus lower. However, even the replacement of CB with 40 wt. % (FA-30/20) results in an acceptable decrease in ΔM, which does not exceed 30%.

The processability of the rubber compounds filled with fly ashes as a carbon black replacement, as represented by the Mooney viscosity values, also improved (decreased viscosity) with higher degrees of replacement (Figure 5).

Based on the above data, the optimum degree of CB with FA replacement selected for further studies was FA-30/20. 

The analysis of grinded ash shows that, despite the fragmentation of the particles, their specific surface area only increased by ca. 15%, remaining at the level of the values characteristic for inactive fillers (max. for FA 2017 samples of approx. 30 m^2^/g (Table 11)). On the other hand, a significant increase in the value of SSA was recorded for the ash fractions (even exceeding 80 m^2^/g (Table 12)); contrary to appearances, not those with the smallest but those with the largest particle size, thanks to the significant development of their surface morphology (Figure 3). The increasing share of quartz and mullite in the coarse-grained ash fraction may be the reason for their greater activity than the fine-grained fraction in rubber compounds. However, the activity of the filler is not reflected in the results of the mechanical tests of rubber vulcanizates (Section 3.2.5), due to the large dimensions of its particles. Such big particles, due to their high mineral contents, could additionally adversely affect the durability of mixers or extruders. Again, because of the better parameters of fly ash A, the influence of fly ash fractionation on the vulcanization kinetics of rubber compounds has been presented only for the example of FA-A from the 2017 filled samples. The compounds containing fractionated ash from 2018 behaved in a similar way. The vulcanization kinetics of rubber mixtures filled with 30 phr of N 220 and 20 phr of fly ash fractions from 2017 are illustrated in Figure 6, and the parameters of their vulcanization are presented in Table 16.

The processability of the rubber compounds filled with CB and different fractions of fly ash as a carbon black replacement, represented by the Mooney viscosity values, are presented in Figure 7.

#### 3.2.2. Filler Dispersion 

Three samples, containing 30 phr of carbon black N 220 and 20 phr of fly ash from 2017 of various fractions (FFA-A < 0.063 mm, 0.063 < FFA-A < 0.125 mm, 0.125 < FFA-A <0.250 mm), were tested (Table 17 and Figure 8). Despite having the most developed surface (BET, CTAB) and the highest content of carbon, both facilitating polymer-filler interactions, the rubber mixtures containing the biggest fly ash fractions were not studied, due to the potential of the big ash particles to initiate crack propagation in rubber vulcanizates. When analyzing the experimental results (Table 17 and Figure 8), it is clear that the coarser the grain, the better the ash dispersion in the mixture. This could be a result of the increase in carbonaceous residues content with the increase in the fraction size. The carbonaceous residues form softer grains than mineral residues, and undergo dispersion much easier during mixing.

#### 3.2.3. Bound Rubber Content (BdR)

The specific surface area value of FA-A 2017 is significantly higher in comparison to FA-B 2018. Keeping in mind that the mechanical requirements for rubber vulcanizates can be met only in the case of partial carbon black (CB) replacement with fly ash, the bound rubber content (BdR) analysis was carried out for rubber compounds containing fractionated fly ash from 2017. The results are presented in Table 18.

The sample with unfractionated fly ash and the sample with the biggest fraction of fly ash show similar bound rubber contents. On the other hand, the samples with the finest grains show a much lower content of BdR. This is generally in line with the SSA value for the individual fraction of the fly ash.

#### 3.2.4. Payne Effect

The Payne effect was studied for the rubber mixes, for which carbon black was partially replaced by FFA (30/20 phr), and is presented in Figure 9.

The changes in G’ with increasing deformation are most pronounced when fine (FFA < 0.063 mm) and medium-size (0.063 < FFA < 0.125 mm) fly ash fractions are used to partially replace CB, which non-directly confirms their worse microdispersion in comparison to the application of the bigger FA fraction studied (0.125 < FFA < 0.250) (Table 19).

This effect seems to be related to the increasing polymer–filler interactions with the increase in FA particle size [33], which agrees with the previous studies on specific surface area, surface energy, dispersion and bound rubber. Generally, for similar compounds, the smaller the Payne effect (smaller ΔG’), the better the filler microdispersion.

#### 3.2.5. Mechanical Properties of Rubber Vulcanizates

The partial replacement of CB with unfractionated fly ash makes rubber vulcanizates softer (Figure 10).

As expected, the partial replacement of CB with fly ash also resulted in an expected, noticeable decrease in the mechanical strength of the rubber vulcanizates (Figure 11).

From an industrial point of view, the depreciation in the mechanical strength of vulcanizates is unacceptable in the case of the complete replacement of carbon black with fly ash at the filling level of 50 phr. However, replacing CB even to 40% (30/20 weight ratio of CB to ash) leads to rubber vulcanizates with still acceptable mechanical strength, from the point of view of technical applications. It is possible that maintaining the relatively high tensile strength of the vulcanizates filled with both CB and the fly ash is caused by an improved CB dispersion resulting from the presence of the fly ash. Similar phenomena can be found in the literature [34]. The partial replacement of CB with fly ash, with the exception of vulcanizates filled only with ash, for which the elongation at break drops dramatically, results in a certain increase in elasticity (Figure 12), which confirms the results of the hardness test (Figure 10).

As expected [35], higher ash contents in the rubber mixture cause the greatest rate of deterioration in the mechanical parameters of the vulcanizates (reduced mechanical strength and hardness). When increasing the amount of ash added to the mixture, an increase in elongation at break was observed, but only until the samples showed sufficient strength. After the samples were completely filled with ash, their elasticity dropped drastically—the samples showed relatively low strength, exhibiting a tendency to crack during elongation. All the mechanical properties determined are summarized in Table 20, where the changes to the mechanical properties of the rubber vulcanizates filled with various amounts of FA in place of CB are compared to those of vulcanizates completely filled with N 220. Apart from the tensile strength (ca.−43%) and reflected elongation at break (−19%), other parameters deteriorated to a lesser extent.

The diagrams below describe the influence of different fractions of fly ash on the mechanical properties of vulcanizates filled with 30 parts of carbon black and 20 parts of fly ash, which was chosen as the most promising system. The sample containing the system with unfractionated ash showed the highest hardness. The samples filled with CB and individual fractions demonstrated similar hardness values, not too dissimilar to those of the CB+FA-All system (Figure 13).

The use of fly ash divided into appropriate fractions has a significant impact on the mechanical properties of rubber vulcanizates. Although the tensile strength and elongation at break for individual fractions are lower in comparison to samples with non-fractionated fly ash, it can be seen that sample CB + 0.063 < FFA < 0.125 yields the best result, not far off from the highest values detected for the former (Figure 14 and Figure 15).

The results of the mechanical tests indicate the competition between grain sizes, which adversely affects the mechanical strength of rubber vulcanizates and their chemical activity, which is higher with bigger grain sizes. It seems unlikely that removing either the smallest or the biggest fractions of fly ash is advisable from the point of view of the mechanical properties of the filled rubber vulcanizates. Alternatively, the reinforcing effect of the non-fractionated FA could be caused by the highest amount of the carbonaceous particles being formed after the burning of the coal. Such particles could exhibit reinforcing potential in a similar way to CB, derived from large SSA values and a high dispersive part of the surface energy. Carbonaceous particles are usually much softer than mineral particles, and therefore could relatively easily break up during rubber mixing, providing satisfactory filler dispersion.

The results of abrasion tests clearly show that the samples filled with CB and fly ash fractions exhibited higher abrasion resistance compared to the sample in which part of the CB was replaced with non-fractionated FA (Figure 16).

It seems likely that rubber abrasion is governed by other laws than mechanical strength and hardness. The bigger size fractions of fly ash showed to be best, for use in a partial replacement of CB, in terms of the abrasion resistance of the rubber vulcanizates studied; however, the medium fractions were not far behind.

## 4. Conclusions

Significant differences in chemical and phase composition between ash samples originating from lignite combusted at the Belchatów Power Plant (Poland), taken from heaps created in different years, have been detected. However, when applying the ash as a co-filler (partially replacing technical CB) in rubber mixtures, these differences become less significant from the point of view of the mechanical properties of the rubber vulcanizates.

Experimental data confirm that fly ash can be a useful and interesting material when subjected to fractionation. Contrary to energy-consuming mechanical grinding, fractionation results in material particles of different chemical and phase compositions, but also of significantly changed specific surface areas, dependent on the size of the material fraction. All this means the fly ash valorized by fractionation can be used as a co-filler in rubber technology. The level of CB replacement, without significant deterioration of the mechanical properties of rubber vulcanizates, was determined to be as much as 40 wt. % in comparison to SBR filled with 50 phr of N220. Fly ash fractions of bigger dimensions exhibit higher specific surface areas in comparison to smaller ones. However, it should be remembered that there is competition for the reinforcement of rubber by fly ash between their particle size and the specific surface area, which is additionally influenced by the phase compositions of filler particles. 

The morphological analysis also revealed a problem with the uneven distribution of ash particles in the rubber. When designing new formulations, it would be necessary to add an agent capable of improving filler dispersion, resulting in its better homogenization with the rubber matrix. Such a role can be enacted by the main filler—in our case, CB. This is why the best results were obtained for rubber mixtures filled with the fly ash fraction with dimensions between 63 and 125 µm, bringing together moderate particle dimensions and a good SSA value.

The division of ash into fractions that differ in chemical and phase composition creates new possibilities in terms of its valorization in comparison to the mechanical crumbling of the material, which is more expensive and not always effective in terms of rubber strengthening. This finding gives an introduction to further research and the chemical modification of a specific ash fraction in order to improve its chemical activity. By taking advantage of the fact that the chemical composition of smaller fly ash fractions facilitates chemical modification, the effect of their smaller SSA can be successfully compensated. The surface modification of different ash fractions, e.g., by silanization, can give new answers as to how one material can produce fillers with different properties that are suitable for various applications.

## Figures and Tables

**Figure 1 materials-15-04869-f001:**
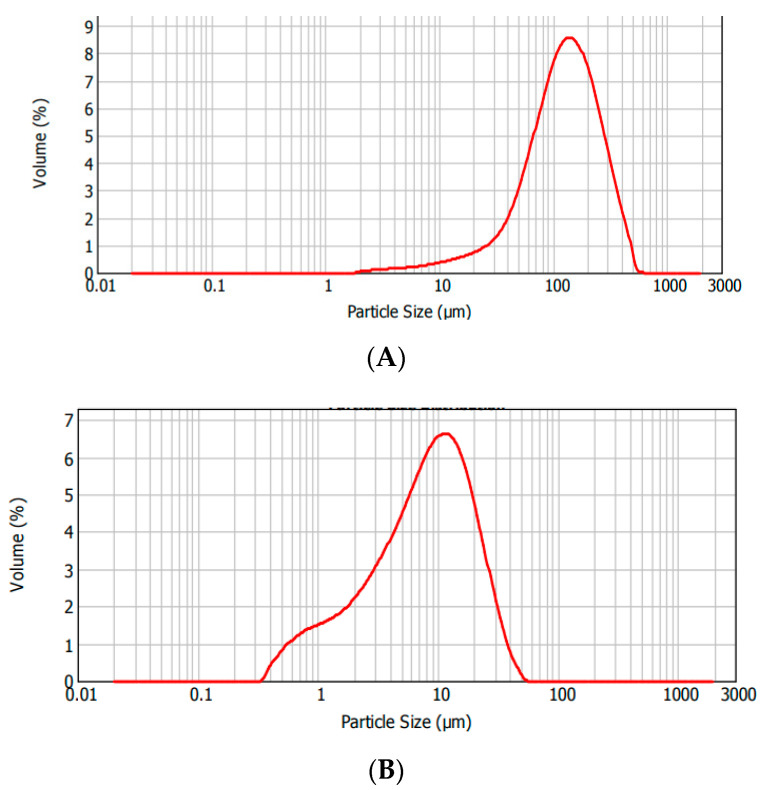
Particle size distribution analysis for fly ash FA-A 2017 samples: (**A**) before grinding, (**B**) after 4 h grinding.

**Figure 2 materials-15-04869-f002:**
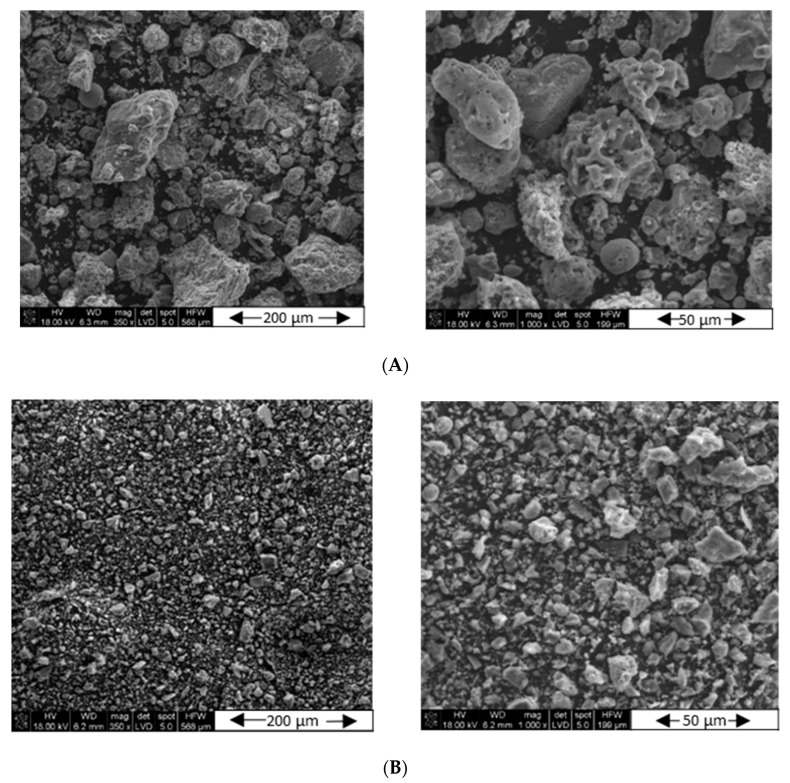
SEM morphology of the fly ash sample FA-A 2017: (**A**) before grinding, (**B**) after 4 h of grinding.

**Figure 3 materials-15-04869-f003:**
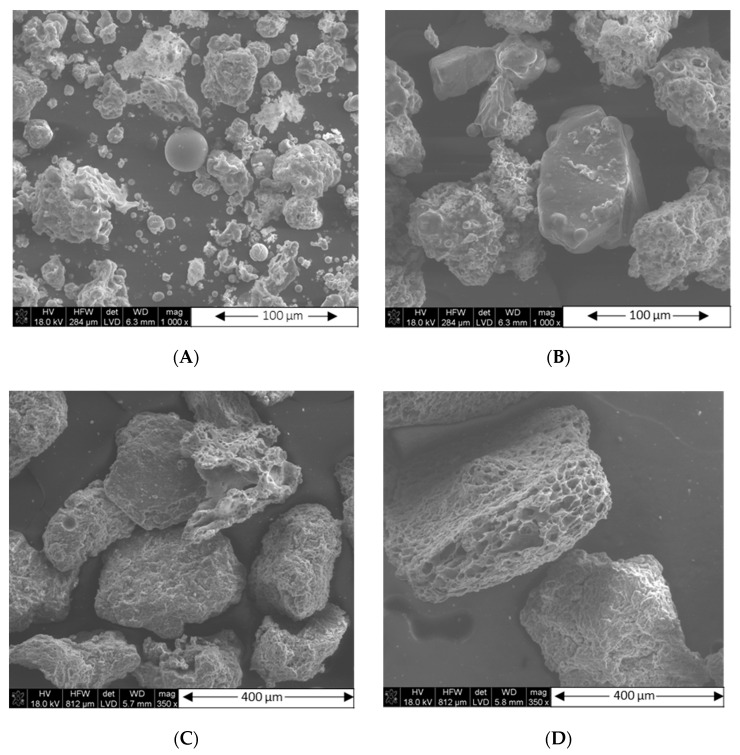
SEM morphology of various fractions of fly ash A from 2017. (**A**) FFA-A 2017< 0.063 mm; (**B**) 0.063 mm < FFA-A 2017 < 0.125 mm; (**C**) 0.125 mm < FFA-A 2017 < 0.250 mm; (**D**) FFA-A 2017 > 0.250 mm.

**Figure 4 materials-15-04869-f004:**
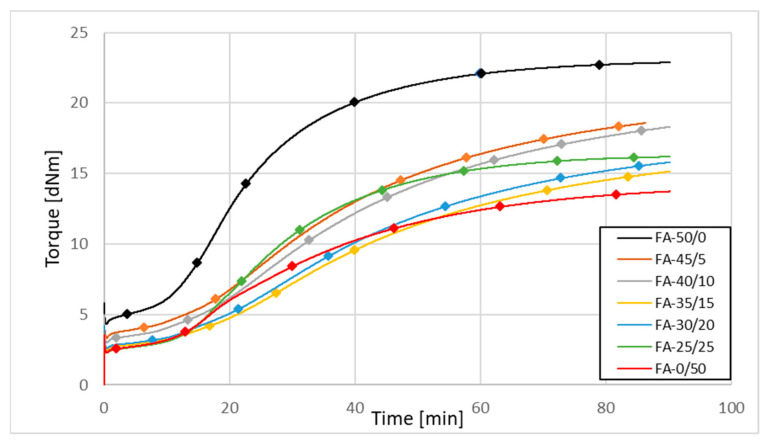
Vulcanization kinetics for rubber mixtures filled with N 220 carbon black partially replaced by fly ash from 2017 (FA-All).

**Figure 5 materials-15-04869-f005:**
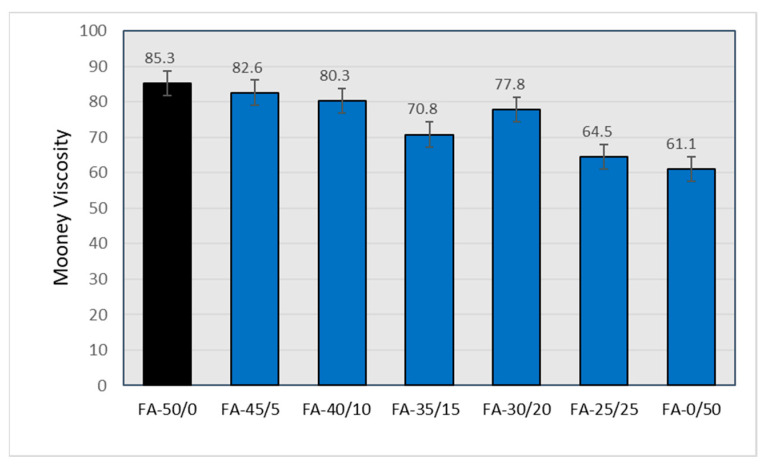
Mooney viscosity for rubber mixtures filled with N 220 and varying amounts of fly ash A (FA-All).

**Figure 6 materials-15-04869-f006:**
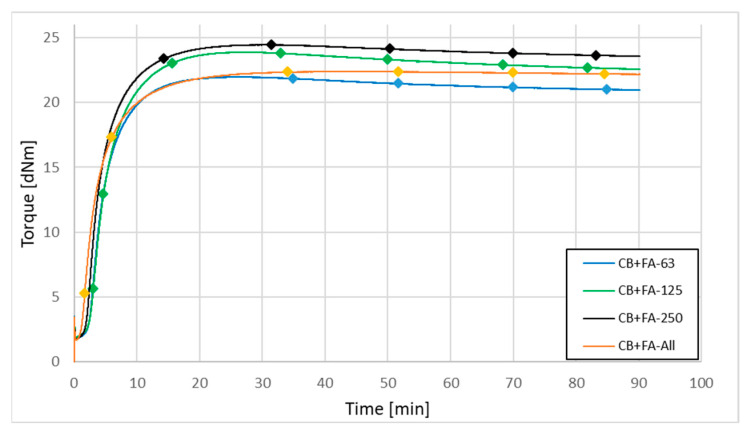
Vulcanization kinetics for rubber mixtures filled with 30 phr of N 220 carbon black and 20 phr of different fractions of fly ash from 2017.

**Figure 7 materials-15-04869-f007:**
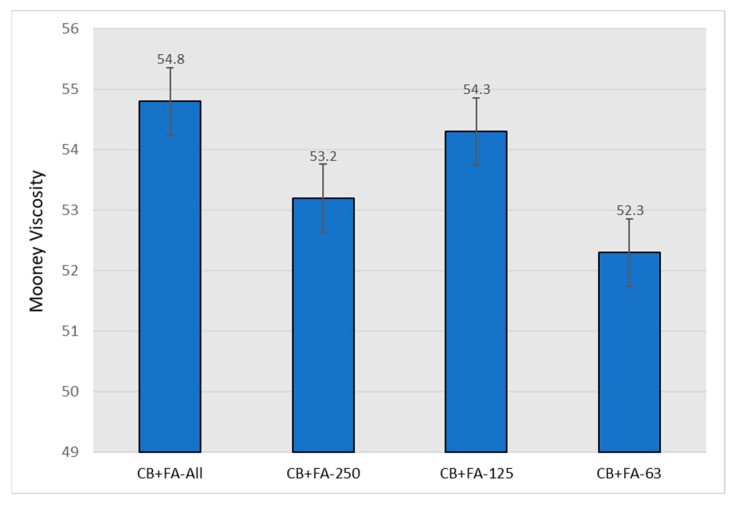
Mooney viscosity of rubber vulcanizates filled with 30 phr of N 220 and 20 phr of different fractions of fly ash from 2017.

**Figure 8 materials-15-04869-f008:**
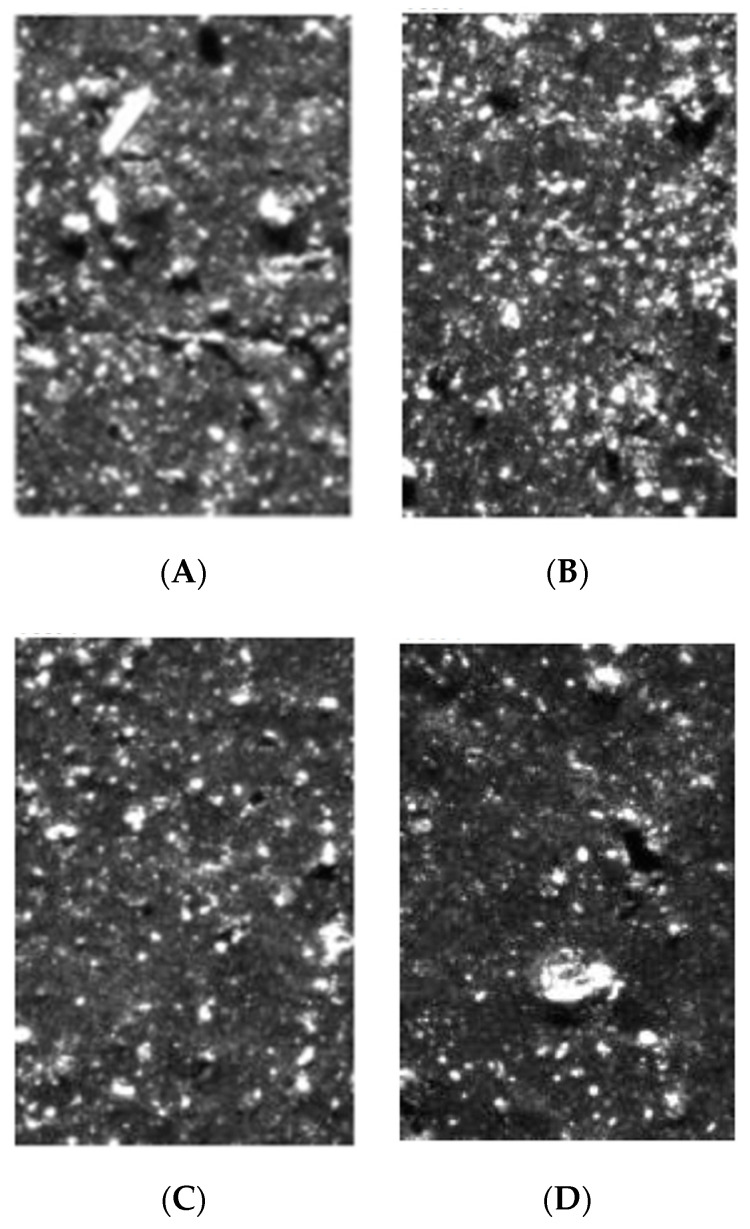
Photos of cross-sections of rubber vulcanizates filled with 30 phr of N 220 and 20 phr of various fractions of fly ash from 2017. (**A**) Compound containing CB+FA-All; (**B**) compound containing CB+FA-63; (**C**) compound containing CB+FA-125; (**D**) compound containing CB+FA-250.

**Figure 9 materials-15-04869-f009:**
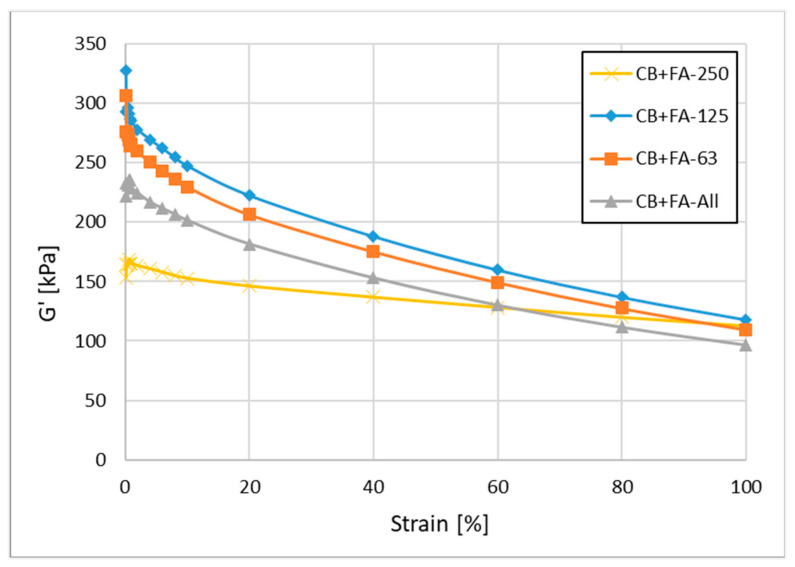
Payne effect in rubber mixtures containing 30 phr of N 220 and 20 phr of non-fractionated and fractionated fly ash from 2017.

**Figure 10 materials-15-04869-f010:**
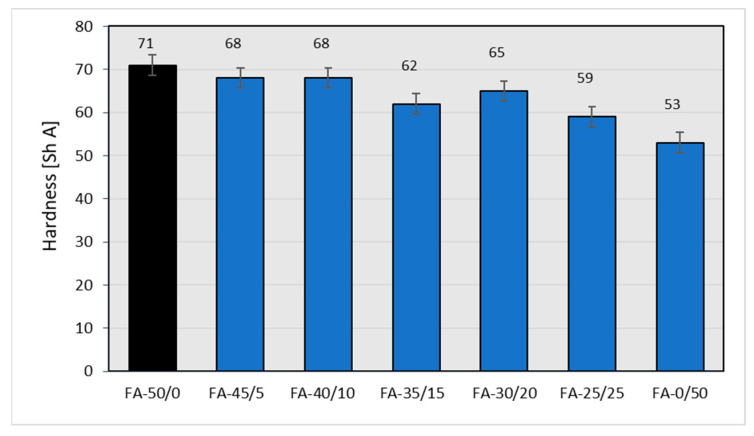
Hardness of rubber vulcanizates filled with N-220 and varying amounts of fly ash from 2017.

**Figure 11 materials-15-04869-f011:**
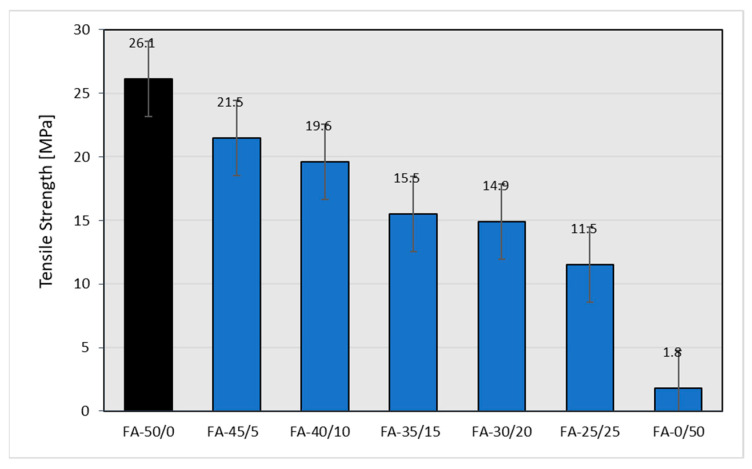
Tensile strength of the rubber vulcanizates filled with N 220 and varying amounts of fly ash from 2017.

**Figure 12 materials-15-04869-f012:**
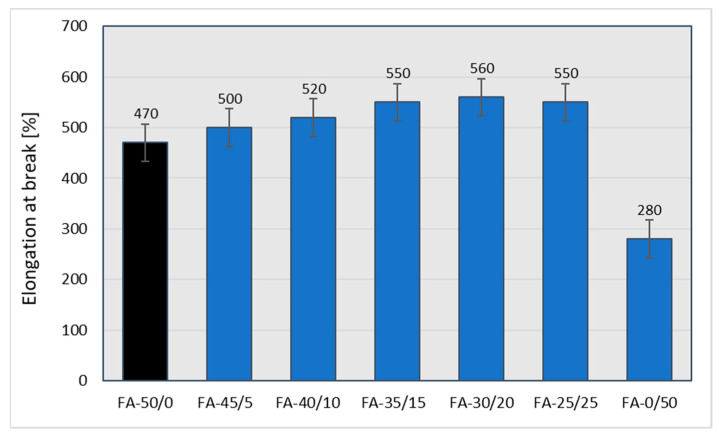
Elongation at break of the rubber vulcanizates filled with N 220 and varying amounts of fly ash from 2017.

**Figure 13 materials-15-04869-f013:**
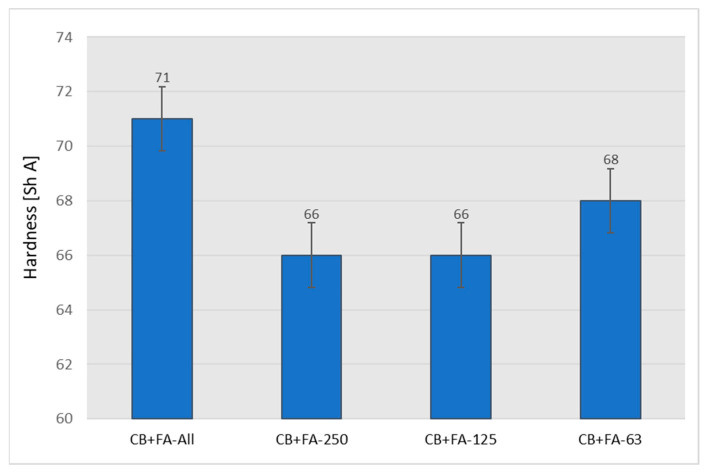
Hardness of the rubber vulcanizates filled with 30 phr of N 220 and 20 phr of the unfractionated or various fractions of fly ash from 2017.

**Figure 14 materials-15-04869-f014:**
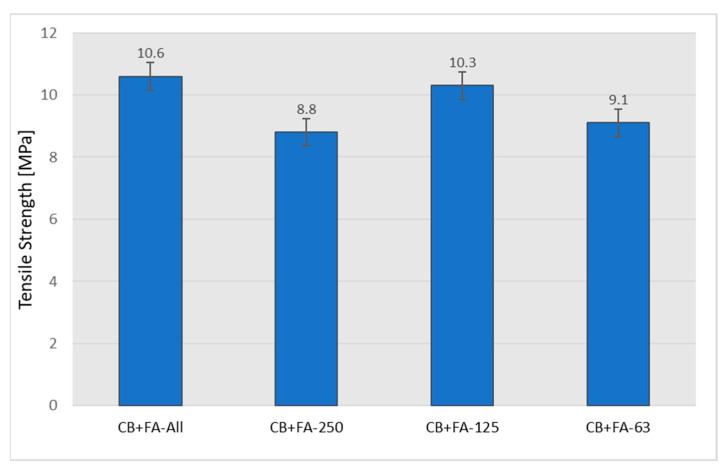
Tensile strength of the rubber vulcanizates filled with 30 phr of N 220 and 20 phr of unfractionated or various fractions of fly ash from 2017.

**Figure 15 materials-15-04869-f015:**
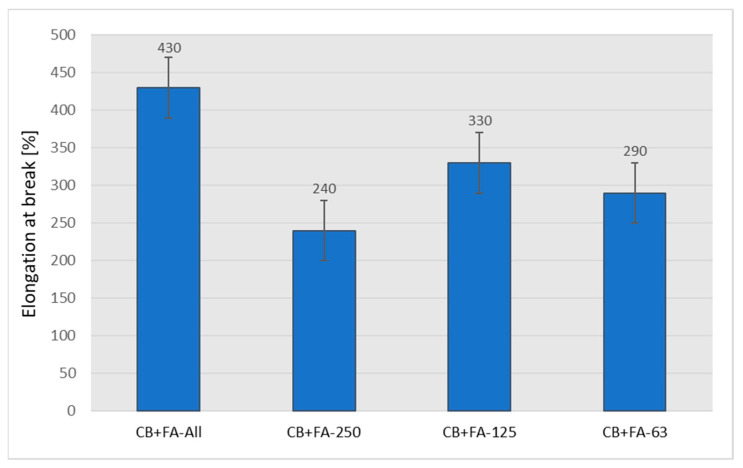
Elongation at break of the rubber vulcanizates filled with 30 phr of N 220 and 20 phr of unfractionated or different fractions of fly ash from 2017.

**Figure 16 materials-15-04869-f016:**
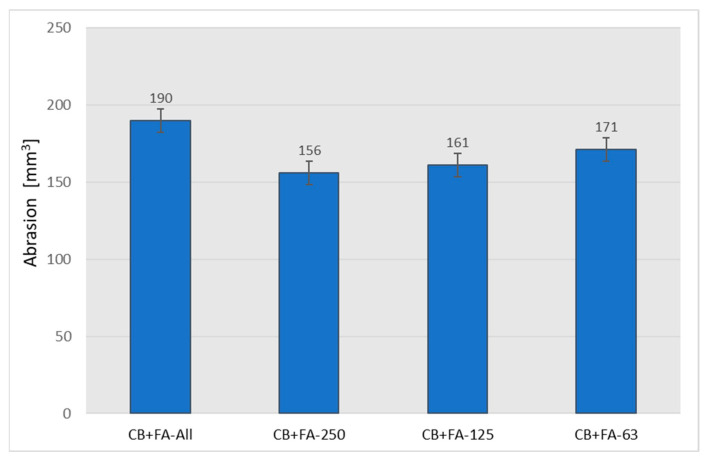
Abrasion of the vulcanizates filled with 30 phr of N 220 and 20 phr of unfractionated or different fractions of fly ash from 2017.

**Table 1 materials-15-04869-t001:** Chemical compositions (wt. %) of ashes from the Belchatow Power Plant, presented in various studies [7,8,9,10].

	*Fly Ash Type [Ref.]*	[7]	[8]	[9]	[10]
*Components*		Limestone Ash A	C-Limestone Ash W	D-Silica Ash V	Limestone Ash
*SiO_2_*		24.0–48.0	64.45	49.38	37.28
*Al_2_O_3_*		15.0–24.0	5.68	25.53	27.46
*Fe_2_O_3_*		3.0–6.0	3.92	5.83	6.78
*CaO*		19.0–39.0	16.50	3.07	19.80
*MgO*		0.5–2.0	3.10	2.22	1.96
*Na_2_O*		0.1–0.4	0.07	0.86	0.14
*K_2_O*		0.1–0.5	0.37	2.70	0.11
*SO_2_*		-	3.87	0.13	-
*Loss on ignition [wt. %]*		-	0.43	8.06	3.28

**Table 2 materials-15-04869-t002:** First step—mixing process for the rubber compounds containing fly ash.

*The Sequence of the Ingredients Addition*	Time from Mixing Start (min) /Temperature (°C)
*Adding of rubber (SBR 1500)*	0/25
*Adding of CB, stearic acid and ZnO*	1.5/90
*Adding of fly ash*	3/100
*End of mixing*	5–6/max. 135

**Table 3 materials-15-04869-t003:** Formulation of the rubber mixes filled with fly ash A and/or CB [phr].

	Rubber Mixes	FA-50/0	FA-45/5	FA-40/10	FA-35/15	FA-30/20	FA-25/25	FA-0/50
Components [phr]								
*SBR, Ker 1500*		100						
*Stearic acid*		1						
*ZnO*		3						
*Carbon black, N 220*		50	45	40	35	30	25	-
*Fly ash A*		-	5	10	15	20	25	50
*Sulfur*		2						
*n-cyclohexyl-2-benzothiazosulfenamide, CBS*		1						

**Table 4 materials-15-04869-t004:** Types of fillers used in rubber mixes.

Range of Filler	Name of Sample
**Native Fly Ash**	FA-All
**FFA < 0.063**	FA-63
**0.063 < FFA < 0.125**	FA-125
**0.125 < FFA < 0.250**	FA-250
**FFA > 0.250**	FA > 250

**Table 5 materials-15-04869-t005:** Formulation of the rubber mixes filled with CB and fly ash A or fly ash A divided into fractions.

	Rubber Mixes	CB+FA-All	CB+FA-250	CB+FA-125	CB+FA-63
Components (phr)					
SBR, Ker 1500		100			
Stearic acid		1			
ZnO		3			
Carbon black, N 220		30			
FA-non fractionated		20			
0.125 < FFA < 0.250		-	20	-	-
0.63 < FFA < 0.125		-	-	20	-
FFA < 0.063		-	-	-	20
Sulfur		2			
n-cyclohexyl-2-benzothiazosulfenamide, CBS		1			
Tetramethylthiuram disulfide, TMTD		1			

**Table 6 materials-15-04869-t006:** Qualitative and quantitative “oxide” analysis of fly ash sample FA-A 2017.

*No.*	*Component*	*Content (wt. %)*
*1*	SiO_2_	49.3
*2*	Al_2_O_3_	24.6
*3*	CaO	12.5
*4*	Fe_2_O_3_	4.68
*5*	TiO_2_	1.17
*6*	SO_3_	1.12
*7*	MgO	0.96
*8*	K_2_O	0.18
*9*	P_2_O_5_	0.12
*10*	SrO	0.05
*11*	BaO	0.04
*12*	Mn_3_O_4_	0.03
*13*	Na_2_O	0.03
*SUM*		99.13
*Loss on Ignition (wt. %)*		4.35

**Table 7 materials-15-04869-t007:** Phase composition of fly ash sample FA-A 2017.

*No.*	*Component*	*Content (wt. %)*
*1*	Silicon dioxide (SiO_2_)	45.7
*2*	Gehlenite (Ca_2_Al_2_SiO_7_)	12.0
*3*	Low albite (NaAlSi_3_O_8_)	26.2
*4*	Mullite (3Al_2_O_3_·2SiO_2_)	15.3

**Table 8 materials-15-04869-t008:** SEM-EDS analysis of the chemical composition of the fly ash FFA-A 2017 fractions in terms of the content of oxides and carbon.

*Grain Size/Fraction*	Σ Oxides (wt. %)	Carbon (wt. %)
*FA-63*	82.7	17.3
*FA-125*	74.7	25.3
*FA-250*	58.8	41.2
*FA > 0.250*	46.9	53.1

**Table 9 materials-15-04869-t009:** “Oxide” analysis of the fly ash FFA-A 2017 fractions.

*Fraction/* *Component*	FA-63 (wt. %)	FA-125 (wt. %)	FA-250 (wt. %)	FA > 0.250 (wt. %)
*SiO_2_*	32.316	33.563	27.121	18.341
*CaO*	19.548	13.555	9.575	7.497
*Al_2_O_3_*	18.638	18.605	14.292	9.424
*Fe_2_O_3_*	6.558	4.814	4.201	3.861
*SO_3_*	2.626	1.385	1.126	3.779
*TiO_2_*	1.267	1.439	1.420	1.557
*MgO*	0.880	0.708	0.477	0.335

**Table 10 materials-15-04869-t010:** The phase compositions of various fractions of the fly ash FFA-A 2017 sample.

*Fraction/* *Component*	FA-63 (wt. %)	FA-125 (wt. %)	FA-250 (wt. %)	FA > 0.250 (wt. %)
*Quartz (SiO_2_)*	30.4	39.9	53.6	73.4
*Akermanite/* *Gehlenite* *Ca_2_Mg(Si_2_O_7_)* *Ca_2_Al[(Si_2_Al)_2_O_7_]*	17.8	7.2	-	1.3
*Albite NaAlSi_3_O_8_*	26.2	22.1	17.3	6.5
*Limestone CaO*	2.2	0.9	-	-
*Calcite CaCO_3_*	2.9	1.5	0.9	0.9
*Hematite α-Fe_2_O_3_*	2.3	1.0	-	-
*Anhydrite CaSO_4_*	3.7	1.1	2.2	3.9
*Brownmillerite* *Ca_2_(Al_2_Fe)_2_O_5_*	3.6	0.9	-	-
*Mullite* *3Al_2_O_3_·2SiO_2_*	10.9	25.5	25.9	14.1

**Table 11 materials-15-04869-t011:** Specific surface area analysis of fly ash FA-A 2017 sample.

*Sample*	CTAB (m^2^/g)	BET (m^2^/g)	BET (m^2^/g) after Grinding
*FA-A*	8.3 ± 0.1	23.7 ± 0.1	27.5 ± 0.1

**Table 12 materials-15-04869-t012:** Results of the BET and CTAB surface area for the fly ash FFA-A 2017 sample divided into fractions.

*Fraction*	BET (m^2^/g)	CTAB (m^2^/g)
*FA-63*	10.7	3.2
*FA-125*	18.2	8.3
*FA-250*	29.5	16.5
*FA > 0.250*	83.6	28.3

**Table 13 materials-15-04869-t013:** Surface energy of fly ash FA-A 2017 sample.

*Surface Energy (mJ/m^2^)*	
*Dispersive component*	33.0
*Polar component*	25.2
*Total value*	58.2

**Table 14 materials-15-04869-t014:** Surface energy of fly ash fractions (FFA) produced by grinding of FA-A 2017.

*Surface Energy*	FA-63 (mJ/m^2^)	FA-125 (mJ/m^2^)	FA-250 (mJ/m^2^)
*Dispersive component*	22.2	32.3	32.7
*Polar component*	7.4	12.3	0.2
*Total value*	29.6	44.6	32.9

**Table 15 materials-15-04869-t015:** Vulcanization parameters of rubber mixtures filled with various amounts of FA-A 2017 (FA-All) and N 220 CB.

	*Parameter*	t_90_ (min)	t_02_ (min)	M_min_ (dNm)	M_max_ (dNm)	ΔM (dNm)
*Sample*						
*FA-50/0*		46.8	10. 7	4.3	22.9	18.6
*FA-45/5*		66.0 [+41.0] *	14.7 [+37.4]	3.3 [−23.3]	18.6 [−18.8]	15.3 [−17.7]
*FA-40/10*		69.7 [+48.9]	15.4 [+43.9]	3.0 [−30.2]	18.3 [−20.1]	15.3 [−17.7]
*FA-35/15*		71.3 [+52.4]	18.5 [+72.9]	2.5 [−41.9]	15.1 [−34.1]	12.6 [−32.3]
*FA-30/20*		70.6 [+50.9]	17.5 [+63.6]	2.6 [−39.5]	15.8 [−31.0]	13.2 [−29.0]
*FA-25/25*		52.7 [+12.6]	14.9 [+39.3]	2.3 [−46.5]	16.2 [−29.3]	13.9 [−25.3]
*FA-0/50*		62.4 [+33.3]	14.9 [+39.3]	2.3 [−46.5]	13.7 [−40.2]	11.4 [−38.7]

* Values in the parenthesis indicate the percentage change of properties compared to the sample FA-50/0.

**Table 16 materials-15-04869-t016:** Vulcanization parameters of rubber mixtures filled with 30 phr of N 220 and 20 phr of different fractions of fly ash from 2017.

	*Parameter*	t_90_ (min)	t_02_ (min)	M_min_ (dNm)	M_max_ (dNm)	ΔM (dNm)
*Sample*						
*CB+FA-All*		10.9	1.4	1.7	22.4	20.7
*CB+FA-63*		10.2	2.5	1.7	22.0	20.3
*CB+FA-125*		11.4	2.6	1.8	23.9	22.1
*CB+FA-250*		10.6	2.1	1.7	24.4	22.7

**Table 17 materials-15-04869-t017:** Filler dispersion analysis for rubber vulcanizates containing 30 phr of N 220 and 20 phr of fractionated fly ash from 2017.

Filler System	Dispersion, D (%)
**CB+FA-All**	50.0
**CB+FA-63**	30.8
**CB+FA-125**	45.5
**CB+FA-250**	60.0

**Table 18 materials-15-04869-t018:** Bound rubber in compounds containing 30 phr of N 220 and 20 phr of fly ash or fractionated FA from 2017.

*Rubber Mix*	Average BdR [%]
*CB+FA-All*	43.7 [39.0–48.3] *
*CB+FA-63*	28.1 [24.5–30.0] *
*CB+FA-125*	33.5 [30.6–36.5] *
*CB+FA-250*	47.2 [44.3–50.2] *

* range of the experimental values.

**Table 19 materials-15-04869-t019:** Payne effect for rubber mixes containing 30 phr of N 220 and 20 phr of non-fractionated and fractionated fly ash from 2017.

	Stress Parameters (kPa)	G′_100%_	G′_max_ − G′_min_
Sample			
**CB+FA-All**		96.8	125.2
**CB+FA-250**		112.9	40.5
**CB+FA-125**		118.0	209.8
**CB+FA-63**		109.6	197.1

**Table 20 materials-15-04869-t020:** Mechanical properties of rubber vulcanizates filled with various amounts of FA from 2017, replacing CB.

	*Parameter*	Mooney Viscosity	Hardness (°Sh A)	Tensile Strength (Mpa)	Elongation at Break (%)
*Sample*					
*FA-50/0*		85.3	71	26.1	470
*FA-45/5*		82.6 [−3.2] *	68 [−4]	21.5 [−17.6]	500 [+6.4]
*FA-40/10*		80.3 [−5.9]	68 [−4]	19.6 [−24.9]	520 [+10.6]
*FA-35/15*		70.8 [−17.0]	62 [−13]	15.5 [−40.6]	550 [+17.0]
*FA-30/20*		77.8 [−8.8]	65 [−8]	14.9 [−42.9]	560 [+19.1]
*FA-25/25*		64.5 [−24.4]	59 [−17]	11.5 [−55.9]	550 [+17.0]
*FA-0/50*		61.1 [−28.4]	53 [−25]	1.8 [−93.1]	280 [−40.4]

* Values in the parenthesis indicate the percentage change of properties compared to the sample FA-50/0.

## Data Availability

The data presented in this manuscript are available on request from the corresponding authors.

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
