# Peer review of "Fly Ash from Lignite Combustion as a Filler for Rubber Mixes. Part I: Physical Valorization of Fly Ash"

_materials, 2022, doi:10.3390/ma15144869_

Round 1
Reviewer 1 Report
In the manuscript entitled “Fly ash from lignite combustion as a filler for rubber mixes. Part I: Physical valorization of fly ash”, the authors present the findings that the addition of classified fly ash as a substitute for carbon black in rubber additives is effective. The data collected in this study contribute to our understanding of this field. However, the structure of the paper needs to be considered more time, and the logic of the paper seems weak due to unnecessary data and figures.
1. Whole paper: there seem to be unnecessarily many figures and tables. For example, is Table 1 necessary?
2. Whole paper: since FA-A 2017 is the only one used for rubber filler, isn't FA-B 2018 data unnecessary for the structure of this paper? (e.g. Fig. 1 b, Fig. 2 b, Fig. 4, Fig. 6, Table 6 b, Table 7 b, Table 8-13)
3. Fig. 12 discusses the dispersion of fly ash filler, but this figure is not clear because there is no scale bar. Also, what does Reference level 1 and Test1, 2 mean?
Author Response
Dear Editor, Dear Reviewers,
We read carefully all comments of the Reviewers and we modified our paper accordingly. We added our answers to the Reviewers’ criticism as insets in his/her comments. For your convenience, all the changes made in the manuscript were made in the review mode and marked with red color.
We appreciate, your time and effort in managing and reviewing this work and we hope that all the comments raised by reviewers were answered in a satisfactory way.
Reviewer 1
In the manuscript entitled “Fly ash from lignite combustion as a filler for rubber mixes. Part I: Physical valorization of flay ash” the Authors present the findings that the addition of classified fly ash as a substitute for carbon black in rubber additives is effective. The data collected in this study contribute to our understanding of this field. However, the structure of the paper needs to be considered more time, and the logic of the paper seem week due to unnecessary data and figures.
- Whole paper: there seem to be unnecessarily many figures and tables. For example, is Table 1 necessary?
We agree with the Reviewer's general remark regarding tables with data for FA-B 2018, which we propose to transfer to Supplementary Materials and therefore change the numbering of the tables. However, we suggest leaving Table 1, which gives information about the different composition of the ashes
- Whole paper: since FA-A 2017 is the only one used for rubber filler isn’t FA-B 2018 data unnecessarily for the structure of this paper? (e.g. Fig. 1 b, Fig. 2 b, Fig. 4, Fig. 6, Table 6 b, Table 7 b, Table 8-13).
In our opinion the work is interesting, among other things, because it allows to compare the behavior of A and B ashes. However, following the Reviewer’s recommendation we moved all the data concerning FA-B 2018 to Supplementary Materials and changed the numbering accordingly.
- Fig. 12 discusses the dispersion of fly ash filler, but this figure is not clear because there is no scale bar. Also what does Reference level 1 and Test 1, 2 mean?
Reference level 1 refers to the degree of dispersion for a mixture with CB only, saved in the device settings - we suggest removing the photo. As for the scale, unfortunately the report from the device generates a photo without a scale, counting only for the number of aggregates. Nevertheless, we suggest to leave one representative photo in each case.
In the name of Authors,
Dariusz Bieliński
Reviewer 2 Report
Dear Authors,
This paper investigates the fly ash as filler for rubber. They have made many studies to characterize the filler and investigated the rubber composites with this filler. The aim of this work is quite interesting. However some major issues should be clarified as suggested below
1) Use the scales for Figure 1 and 2.
2) Why you use sometimes single accelerator and binary accelerators for different rubber composites? In this case, it is needed to study curing activity of 50 phr CB and 50 phr FA-all separately with the binary accelerators system.
3) Curing studies are mainly done on shear mode whereas tensile mechanical properties are studied on tensile mode. Bigger particle size can improve Δ torque due to filler anisotropy but it does not provide better tensile properties due to weak filler-polymer interactions. Binary filler sometimes undergo mutual filler dispersion and provide better tensile properties. This paper could be helpful for discussing the mechanical properties (Polymer Bulletin (2022), Vol. 79, pages 2707–2724).
4) Mechanical modulus at different elongations can be discussed if there may some advantages in the binary filler systems.
Author Response
Dear Editor, Dear Reviewers,
We read carefully all comments of the Reviewers and we modified our paper accordingly. We added our answers to the Reviewers’ criticism as insets in his/her comments. For your convenience, all the changes made in the manuscript were made in the review mode and marked with red color.
We appreciate, your time and effort in managing and reviewing this work and we hope that all the comments raised by reviewers were answered in a satisfactory way.
Reviewer 2
This paper investigates the fly ash as filler for rubber. They have made many studies to characterize the filler and investigated the rubber composites with this filler. The aim of this work is quite interesting. However, some major issues should be clarified as suggested below.
- Use the scaled for Figure 1 and 2.
The marking of the X and Y axes in Figures 1 and 2 has been completed. X refers to particle size [µm], whereas Y refers to volume [%].
- Why you use sometimes single accelerator and binary accelerators for different rubber composites? In this case, it is needed to study curing activity of 50 phr CB and FA-All separately with the binary accelerators system.
The idea behind the work was that we first compare vulcanizates with different ashes (A and B) and create mixtures in different CB / FA ratios. The results of the vulcanization kinetics show that for the S + CBS system we have very long t2 and t90 times. Therefore, we decided to add TMTD to reduce vulcanization time, making the process commercially viable.
- Curing studies are mainly done on shear mode whereas tensile mechanical properties are studied on tensile mode. Bigger particle size can improve Δ torque due to filler anisotropy but it does not provide better tensile properties due to week polymer-filler interactions. Binary filler sometimes undergo mutual filler dispersion and provide better tensile properties. This paper could be helpful for discussing the mechanical properties (Polymer Bulletin 2022, Vol. 79, pages 2707-2724).
Thank you for this comment. Based on it, we have added a complementary comment, citing the relevant literature reference. “It is possible that maintaining the relatively high tensile strength of the vulcanizates filled with both CB and the fly ash is caused by an improved CB dispersion resulting from the presence of the fly ash. Similar phenomena can be found in the literature [30].”
- Mechanical modulus at different elongations can be discussed if there may some advantages in the binary filler systems.
The Reviewer is generally right, but the aim of our research was not to show that ash can replace CB in 100%. It was known from the beginning that this was not possible, so we tried to test compounds with different CB / FA ratios to determine how the ash content affects specific rubber parameters. This is an introduction to recommending the proportions in which CB can be partially replaced with ash without significantly deteriorating the mechanical properties of the rubber.
In the name of Authors,
Dariusz M. Bieliński
Reviewer 3 Report
The paper describes the potential application of FA as a filler for rubber mixes.
There is a typo in the paper.
Why were only 2 years of FA analysis selected? Given the difference in composition, it would be interesting to see at least another 2 years.
And another thing: it would be good to show the results for FA - 2018, although the authors say that it behaves similarly to the FA from 2017.
The authors do not compare their results with the results from the literature. Are they similar? Better?
Author Response
Dear Editor, Dear Reviewers,
We read carefully all comments of the Reviewers and we modified our paper accordingly. We added our answers to the Reviewers’ criticism as insets in his/her comments. For your convenience, all the changes made in the manuscript were made in the review mode and marked with red color.
We appreciate, your time and effort in managing and reviewing this work and we hope that all the comments raised by reviewers were answered in a satisfactory way.
Reviewer 3
The paper describes the potential application of FA as a filler for rubber mixes.
- There is a typo in the paper.
We checked the article for typos and made corrections.
- Why were only 2 years of FA analysis selected? Given the difference in composition, it would be interesting to see at least another 2 years.
In fact, we checked the behavior of three ashes from the following years 2017-2019. They behaved in a similar way in rubber mixtures. In the publication, we focused on the one for which we obtained the best results.
- And another thing: it would be good to show the results for FA - 2018, although the authors say that it behaves similarly to the FA from 2017.
According to other reviewers, the article is too extensive. Following their suggestion, some of the results were moved to Supplementary Materials. Also in our opinion, publishing another, similar results would not bring anything new to the essence of the work.
- The authors do not compare their results with the results from the literature. Are they similar? Better?
The aim of our research was not to show that CB can be partially replaced with ash without significantly deteriorating the mechanical properties of the rubber. For this reason, our final results with regard to the mechanical properties of rubber with ash are similar to those available in the literature. However, the main achievement of the article was to show that ash can be used for this purpose, not necessarily ground with a large energy input, but fractionated on a sieve. The individual sieve fractions differ not only in size, but also in their chemical and phase composition, which can be further used for their reinforcing properties in relation to rubber. This type of information has not been described in the literature so far.
In the name of Authors,
Dariusz M. Bieliński
Round 2
Reviewer 1 Report
The correction has been completed and improved in response to the points. It could be accepted in present form.
Author Response
On behalf of the Authors, I would like to thank the Reviewer for valuable commentsand favorable opinion. Dariusz Bieliński
Reviewer 2 Report
I appriciate your revision.
Author Response

(The authors gave the same response as above.)

Reviewer 3 Report
-
Author Response
I cannot see any detailed comments or suggestions from the Reviewer. Therefore, Iassume that he accepted the article in its current form.
Dariusz Bieliński